# Possibilities of Detecting Damage Due to Osmosis of GFRP Composites Used in Marine Applications

**Waldemar Swiderski \*** and **Martyna Strag**

Military Institute of Armament Technology, Prymasa Stefana Wyszynskiego 7 St., 05-220 Zielonka, Poland
\* Correspondence: waldemar.swiderski@wp.pl; Tel.: +48-509-420-225

**Abstract:** The marine composites market is driven by the increasing demand for lightweight, corrosion-resistant, and impact-resistant boats. Polymer matrix composites are currently the most popular composite material in marine applications. Fiberglass composites are practically the main type of fiber composites that are used extensively in marine applications. Due to the aggressive sea environment, composite structural elements of ships are exposed to damage due to the phenomenon of osmosis. This damage is also favored by defects that result from impacts and technological errors during the production of these elements. Non-destructive testing methods are necessary to detect damage in the internal structure of the composite. The paper presents a numerical analysis of the possibility of using vibrothermography in the detection of defects in glass–fiber reinforced laminates in marine applications. Numerical simulations have shown that the most favorable method for detecting defects will be acoustic waves. This is an unusual application because, as a rule, the range of ultrasonic waves is used in vibrothermography. In our further works, it is planned to verify numerical calculations through experimental research. The applicability of the terahertz technique was also assessed. During the experimental testing, all defects in the test sample of the glass–fiber reinforced composite were detected using this technique. The presented results indicate the applicability of the presented methods for the detection of defects in composites used in marine applications.

**Keywords:** composites; vibrothermography; terahertz radiation; non-destructive testing

## 1. Introduction

Interest in composite materials in many industries results from the possibility of designing their functional properties required in a specific application. This creates the possibility of new design solutions that have been previously unavailable with the use of traditional materials, e.g., metals. Fiber-reinforced composites are particularly attractive materials. The use of fibers in the composite makes the material stronger and stiffer [1]. It may have physical parameters comparable to metals but have a lower specific weight. Most often, composites employ textile materials joined together with plastics, such as a binder, to create multi-layered composite materials used, for example, in marine applications. Glass–fiber reinforced composites are most often used in the construction of yachts. More and more often there are constructions where there is a layer of rigid polyvinyl chloride foam between two layers of glass–fiber reinforced composite X.

Damage to the composite structure may occur both as a result of technological errors in the production phase and during operation as material fatigue. The main type of damage to watercraft is damage to the composite structure caused by impacts, which is one of the most critical failures. The effects of impact damage can be considered as the internal fracture surface is characterized

(1) impact damage resistance which is related to the response and damage caused by impact [2],
(2) impact damage tolerance, associated with the reduced stability and strength of the structure due to the damage [3,4].

Another factor that contributes to the damage to the composite hulls of watercraft is the phenomenon of osmosis (Figure 1). Osmosis in hulls is the penetration of water through the gelcoat layer into the laminate. Even the best-made laminate is not a homogeneous structure. There are micro-air bubbles and microcracks, both in the resin itself and at the junction of resin and glass. In these voids of the laminate, water collects, which forms acetic and hydrochloric acids and glycol as a result of hydrolysis. Since glycol is a strongly hygroscopic substance, the amount of water absorbed into the hull increases and the degradation process accelerates. The molecules of water contaminated with compounds formed as a result of hydrolysis strive to equalize the degree of their contamination with the environment. As a result of diffusion, pure water molecules penetrate the laminate unhindered, accelerating the osmosis phenomenon. Hydrolysis and diffusion are more intense in seawater, which is alkaline (pH 8–8.5). As a result of these phenomena, the connections between the fibers of the glass reinforcement and resins were destroyed in the first place. Water spreads along the reinforcement fibers, separating them from each other. The effect of this phenomenon is the gradual degradation of the laminate manifested by the formation of blisters filled with a specifically fragrant, acidic liquid with a pH of 0 to 6.5. Osmotic blisters can vary in size—from the size of a pinhead to a diameter of 10 cm. As a result of osmosis, not only does the laminate structure gradually deteriorate but the water content in the laminate increases, and the weight of the hull increases. Changing the water content in the laminate has a direct impact on the depth of immersion, maneuverability, speed, and fuel consumption [5].

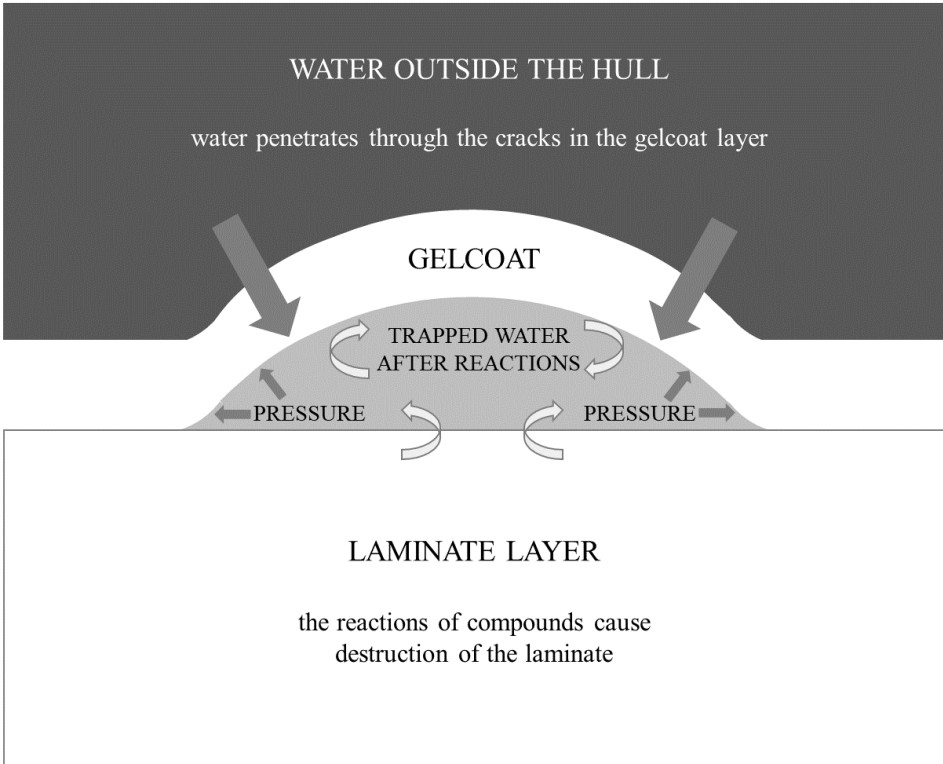

**Figure 1.** The process of the osmosis phenomenon [5].

Damage resulting from impacts or technological errors in the production of composite hulls of floating units accelerate the formation of damage caused by the phenomenon of osmosis. Figure 2 shows osmosis foci on the surface of the yacht's hull after removing the gelcoat layer. In this picture you can see what damage can be caused by the phenomenon of osmosis.

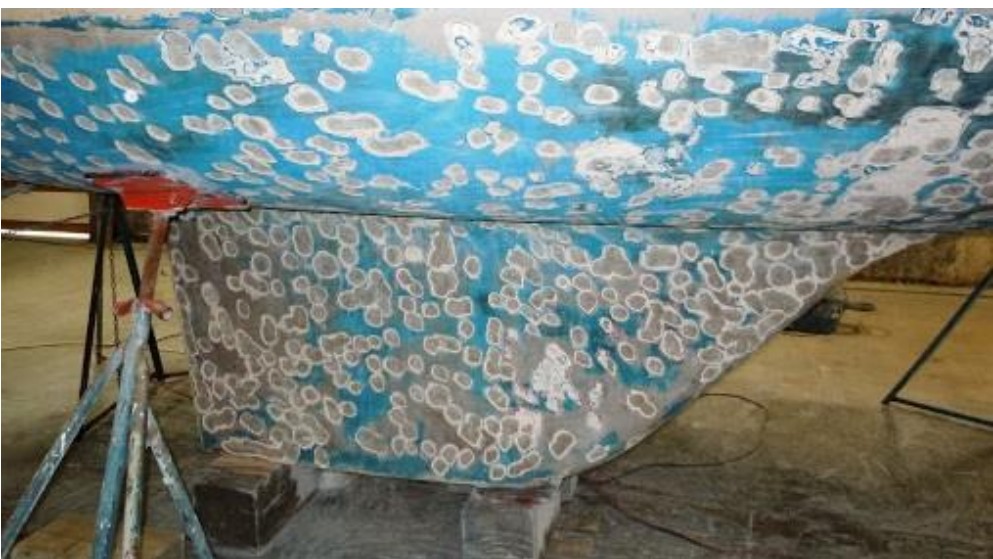

**Figure 2.** Osmosis foci on the yacht's hull [5].

So far, a commonly used method of detecting the effects of osmosis has been tapping, for example, with a hammer on the surface of the yacht's hull. The characteristic sound caused by tapping is information about the existence of damage due to osmosis. However, such a damage detection technique requires the repairperson to have a lot of experience, especially when the damage surface is very small. This method, in many cases, does not allow us to detect the places where the osmosis process begins.

Compared to the literature on non-destructive testing of composites, publications on testing composites used in marine applications are a small part. More often, it is only mentioned that composites used in marine applications can be tested with the presented NDT methods [6]. In the study of composites in marine applications, in most cases, the methods used in similar composites for other purposes are used [7]. These are methods such as ultrasonic tests, optical methods (holography and sherography [8]), microwaves, thermography, and radiography [9]. A certain exception is the acoustic method, which is often used in the study of composites used in marine applications [10], and occasionally in other applications. In recent years, results from the use of terahertz radiation have been presented, most often from the use of THz SWT (Stationary Wavelet Transform) [11] and THz TDS (Time–Domain Spectroscopy) [12] methods.

Non-destructive testing methods make it possible to detect the defects described above, formed in composite hulls of watercraft. The purpose of our work was to analyze the possibility of determining the parameters of non-destructive testing method, which would be both effective in detecting defects resulting from the phenomenon of osmosis and would allow relatively fast testing of the composite hull. Our many years of experience in non-destructive composites showed that one of these such methods is vibrothermography, and the possibilities of its use for detecting defects in composite hulls will be analyzed in this paper. The possibility of using the terahertz transmission method to detect defects caused by osmosis is also analyzed.

## 2. Vibrothermography

In infrared thermography, the distribution of the temperature field on the surface of the test object is used to assess the condition of the test object. Modern thermal imaging cameras can record changes in electromagnetic energy in the infrared band radiated from the surface of the tested object with high frequency and temperature resolution (approx. 15 mK). The whole process, after transforming the radiated energy into an electronic signal, is recorded in the form of a sequence of images [13].

Non-destructive testing with infrared thermography can be performed with both passive and active techniques [14–22]. When using an active technique, it is necessary to use the source of thermal excitation of the tested object. For this purpose, either the heating or the cooling source of the test object can be used. Material defects which, before the test begins, have a homogeneous temperature equal to the ambient temperature do not generate "useful" temperature signals and, for this purpose, require heating or cooling the whole object or its part. During such a test a dynamic temperature field is created, and the results of the test of temperature distribution depending on the observation time.

The name vibrothermography itself can be slightly misleading because this method does not rely on a direct combination of the physical phenomena used in the vibration method and thermography. The vibration method uses information on the change of direction and time of passage of waves inside the tested structure, which are affected by the elasticity and homogeneity of this structure. The detection of defects is influenced by, among other factors, the length of the mechanical wavelength and the ratio of the wavelength to the size of the defect. Only the source-generating mechanical waves were adopted from the vibration method to vibrothermography.

In vibrothermography, the evaluation of latent structural heterogeneities of materials is based on changes in the surface temperature field under cyclic mechanical loads. The reason for the increase in temperature is the internal friction of the cavity walls when they are stimulated by mechanical waves. If the cyclic loads do not exceed the elasticity of the material and the rate of their changes is high, then heat losses due to thermal conductivity are small. After removing the load, the tested object returns to its original shape and temperature. Thermographic methods are usually non-contact methods. Vibrothermography differs from this principle, and the generator of mechanical waves must be in contact with the surface of the tested object.

In order to effectively detect defects using vibrotermography, the most favorable values of the frequency and amplitude of the mechanical waves for thermal excitation of the tested object should be determined. The ThermoSon software was developed for this purpose.

In vibrothermography, ultrasounds are usually used to excite the material under test, the frequency range of which is from 20 kHz to 1 GHz. Occasionally, sound waves with a frequency range of 20 Hz to 20 kHz are used.

## 3. Modelling Vibrothermography NDT

### 3.1. Matematical Problem

The principles of vibrothermography are illustrated in Figure 3.

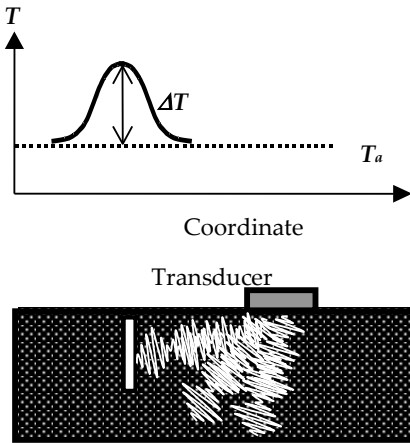

**Figure 3.** Vibrothermography principle [21].

Theoretical numerical calculations can be divided into two steps with 3D solutions to the following problems:

- propagation of mechanical waves in the object and the thermal power generated by the defect,
- propagation of heat generated by the defects based on the results of the first stage.

Below, a description of the steps for numerical calculations is presented. The presented algorithms were developed on the basis of the theoretical approach described in the publications [23,24].

The thermomechanical model implements parallel numerical solutions for the propagation of thermal energy and mechanical waves. Figure 3 shows the mechanical stresses (normal and tangential) that act on each face of the elementary volume of the tested object. Along the two coordinates, the tangential stresses have two components. By adding up one normal component and two tangent components there are three stress components on each surface of the parallelepiped. For example, on the surface perpendicular to X we have one normal component, $\sigma_x$, and two tangent components, $\tau_{xy}$ and $\tau_{xz}$ (the first subscript shows the coordinate axis which is parallel to the normal vector of the parallelepiped surface, and the second subscript indicates the coordinate axis which is parallel to the tangent stress component).

As described in Figure 4, the mechanical stresses acting on each plane of a solid parallelepiped act as three components of stress. For example, on the plane perpendicular to the $x$ coordinate, these are $\sigma_x$, $\tau_{xy}$, and $\tau_{xz}$ stresses. The balance of forces along $x$ can be described by the formula:

$$(\sigma_{x+0} - \sigma_{x-0})\Delta y \Delta z + (\tau_{y+0,x} - \tau_{y-0,z})\Delta x \Delta z + (\tau_{z+0,x} - \tau_{z-0,x})\Delta x \Delta y + X \Delta x \Delta y \Delta z = 0 \tag{1}$$

where $\Delta x$, $\Delta y$, and $\Delta z$ are the parallelepiped dimensions and $X$ is the projection of volume forces (if they are present) on the $x$ axis. The subscripts $x + 0$ and $x - 0$ correspond to the maximum and minimum $x$ coordinates.

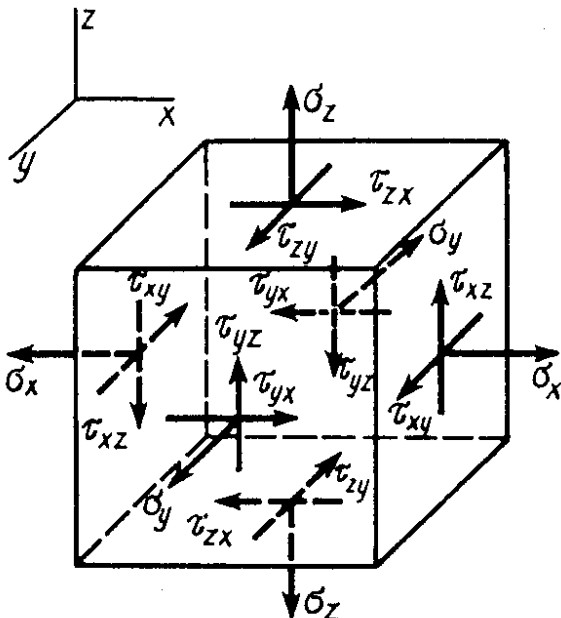

**Figure 4.** Distribution of stresses acting on the tested sample with an elemental volume [25].

The differential form of Equation (1) will be as follows:

$$\frac{\partial \sigma_x}{\partial x} + \frac{\partial \sigma_{yz}}{\partial y} + \frac{\partial \sigma_{zx}}{\partial z} + X = 0 \tag{2}$$

The numerical solution of the obtained set of basic equations was made using ThermoSon software:

$$(\lambda + 2G)\frac{\partial^2 U}{\partial x^2} + G\left(\frac{\partial^2 U}{\partial y^2} + \frac{\partial^2 U}{\partial z^2}\right) + (\lambda + G)\frac{\partial^2 V}{\partial x \partial y} + (\lambda + G)\frac{\partial^2 W}{\partial x \partial z} = 0 \tag{3}$$

where $\lambda$ is the Lamė constant, $G$ is the shear modulus, and $U$, $V$, and $W$ are the displacement projections on the $x$, $y$, $z$ coordinates.

The right element of the Lame Equation (3) was replaced with the corresponding components of the inertia forces $F_{in}$. This is because we are considering elastic waves in a solid.

$F_{in}$ in a chosen elementary volume. For instance, in the $x$ direction:

$$F_{in,x} = \rho \frac{\partial^2 U}{\partial \tau^2} \tag{4}$$

where $\rho$ is the material density.

The following model boundary conditions were adopted:

(1) normal and tangential stresses on the surface of the upper and lower plate do not occur
(2) zero vertical displacements on the lower surface and forced harmonic vibrations at the point of stimulation.

The internal fracture surface is characterized by nonlinear boundary conditions. At the crack compression stage, no tangential stresses are assumed, and at the crack deformation stage, the condition of zero normal stresses is added. Assuming that the appropriate elements of Equation (3) are equal to zero, the above conditions will be achieved. As a starting condition, the displacements are assumed to be zero at $\tau = 0$. By solving Equation (3), it is possible to calculate the displacement for the $x$, $y$ and $z$ coordinates at each time step. The vertical displacement of the surface at the stimulation point is described by the equation:

$$W = A[1 \Delta cos(2\pi f \tau)]/2 \tag{5}$$

where $A$ and $f$ are the vibration amplitude and frequency, respectively.

The heating power in a crack perpendicular to the $x$ direction is calculated by:

$$P = \frac{k_{fr}\sigma_x S_{fr}}{\tau^*} \int_0^{\tau^*} \left|\frac{\partial U}{\partial \tau}\right| d\tau \tag{6}$$

where $k_{fr}$ is the crack wall friction coefficient, $\sigma_x$ is the stress normal to the crack surface, and $S_{fr}$ is the crack surface.

For the second step, a 3D heat conduction equation was numerically solved to calculate the temperature distribution in a sample [16].

### 3.2. Model

The multi-layered structure to be tested consists of layers of glass fabric joined with an epoxy resin glue. The analysed model composite has a thickness of 10 mm and lateral size 250 mm. Three air-filled defects with dimensions D1—50 × 100 mm, D2—20 × 50 mm, and D3—20 × 50 mm (Figure 5) were placed at different depths (D1—0.25 mm, D2—0.5 mm, and D3—0.25 mm) below the surface of the tested composite sample model. All defects are very thin and have a thickness of 0.1 mm.

The model takes into account differences in the propagation of heat generated by the defect as a result of simulation with mechanical waves, which are when heat propagates along the fibers and perpendicular to them. The structure of the model corresponds to the structure of the real composite used for the hulls of the boats.

Table 1 contains the thermal and strength parameters that were determined experimentally. The thermal properties of air were adopted on the basis of the literature (Table 2) [14].

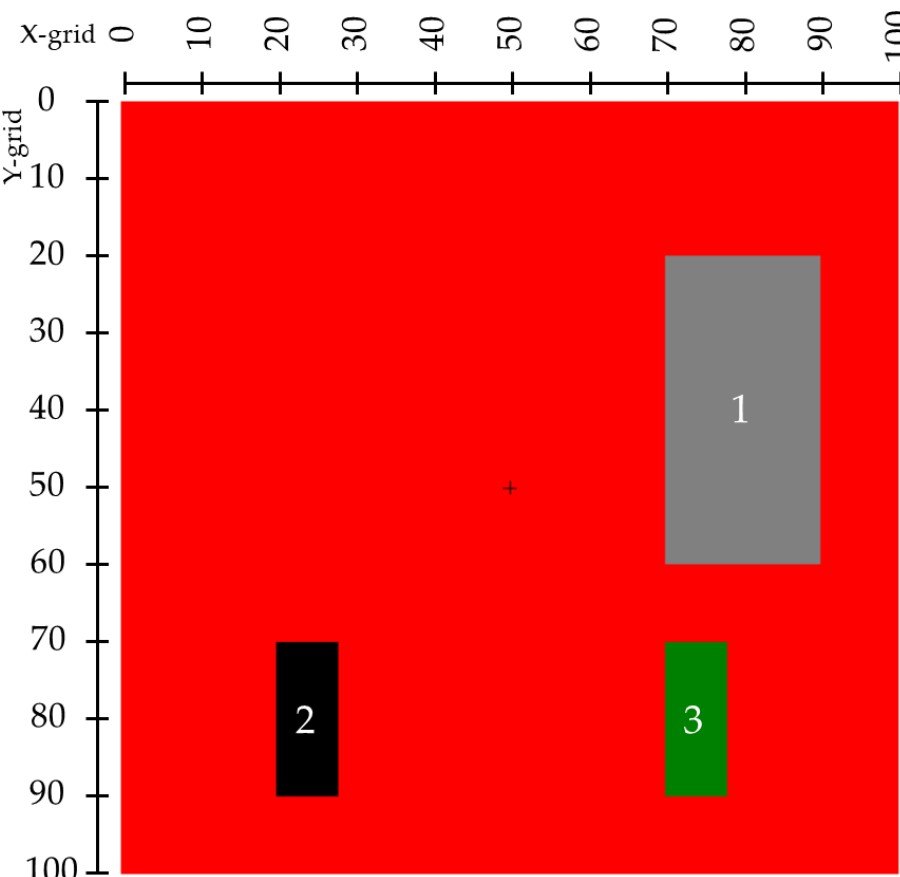

**Figure 5.** Model of composite where 1, 2, and 3 correspond to the defects 1, 2, and 3, respectively.

**Table 1.** Thermal and strength properties of the composite [26].

| Parameter | Value | Unit |
|---|---|---|
| Specific heat | 1.62 | $J \cdot kg^{-1} \cdot K^{-1}$ |
| Density | 2000 | $kg \cdot m^{-3}$ |
| Thermal conductivity | 0.38 | $W \cdot m^{-1} \cdot K^{-1}$ |
| Poisson's coefficient | 0.24 | - |
| Young's modulus | 36 | GPa |

**Table 2.** Thermal properties of air.

| Parameter | Value | Unit |
|---|---|---|
| Specific heat | 1005 | $J \cdot kg^{-1} \cdot K^{-1}$ |
| Density | 1.2 | $kg \cdot m^{-3}$ |
| Thermal conductivity | 0.07 | $W \cdot m^{-1} \cdot K^{-1}$ |

## 4. Results and Discussion

The performed numerical calculations showed that the best imaging of all defects was obtained at the frequency f = 1 kHz (this is the frequency of the sound wave) and the amplitude A = 0.1 mm. Figure 6 shows the graphs of temperature changes over time on the sample surface over the defects. The simulation time of generating vibrations (heating) was $t_h$ = 3 s, and the total simulation time t = 5 s. The point of application of the center of the vibration transducer (with an area of 1 cm$^2$) on the sample surface was: x = 50, y = 50 (Figure 5).

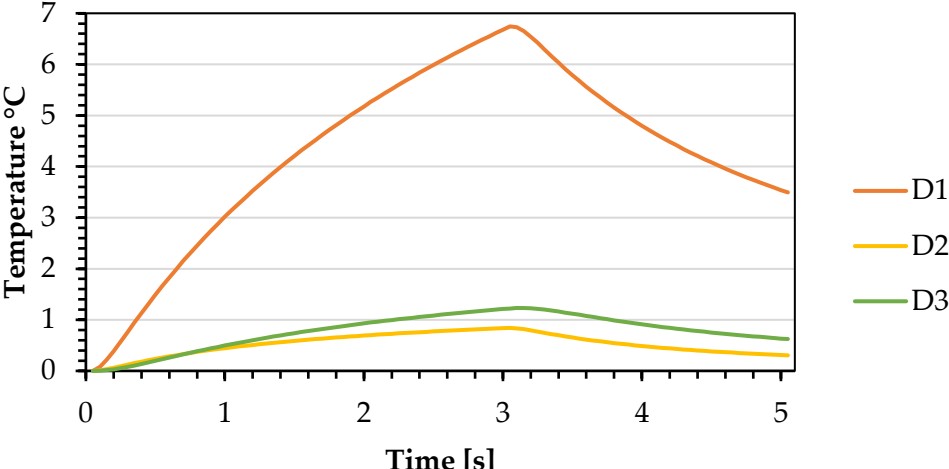

**Figure 6.** The course of temperature changes on the sample surface over the defects (location of the center of the vibration transducer x = 50, y = 50).

Figure 7 shows the temperature distribution after 3 s from the start of vibration generation. Then, as shown in Figure 5, the increases in temperature values above the defects were the highest.

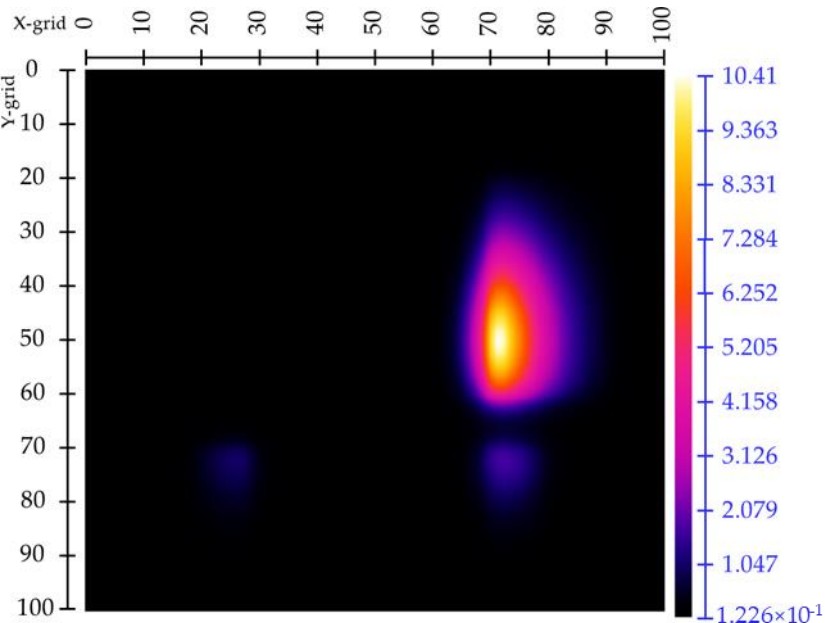

**Figure 7.** The temperature distribution after 3 s from the start of vibration generation (location of the center of the vibration transducer x = 50, y = 50).

The shape of defects cannot be determined in Figure 7, which shows the thermogram simulation. The Fourier transform was used to improve this visualization (Figure 8). Such an operation is possible thanks to the ThermoSon software used in the simulation. The principle of processing thermograms using the Fourier transform is described in the paper [27]. As you can see in Figure 8, the shape of the defects is very visible and consistent with the model shown in Figure 5. The image shown in Figure 5 was obtained for the 14th harmonic.

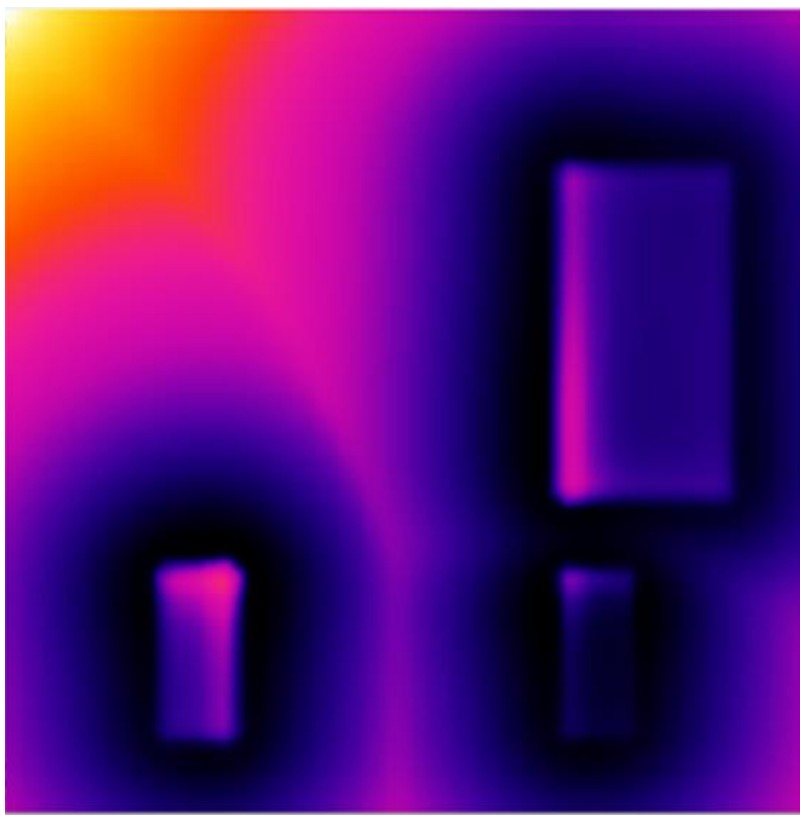

**Figure 8.** Image of the thermogram from Figure 6 after analysis using the Fourier transform for the 14th harmonic.

Determining the shape and dimensions of the defect, especially when it is not visually visible, is important to repair the damaged part of the hull of the watercraft. The example of using the Fourier transform presented in the article shows how helpful the thermogram processing methods (described in [27]) can be in detecting defects using infrared thermography. Figures 9 and 10 show how the temperature rise changes with time during the heating and cooling phase over the defects when the position of the center of the vibration transducer changes.

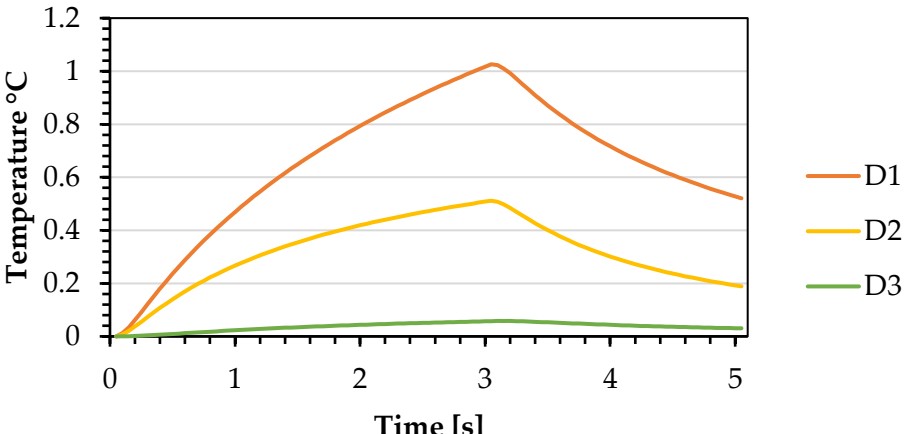

**Figure 9.** The course of temperature changes on the sample surface over the defects (location of the center of the vibration transducer x = 40, y = 50).

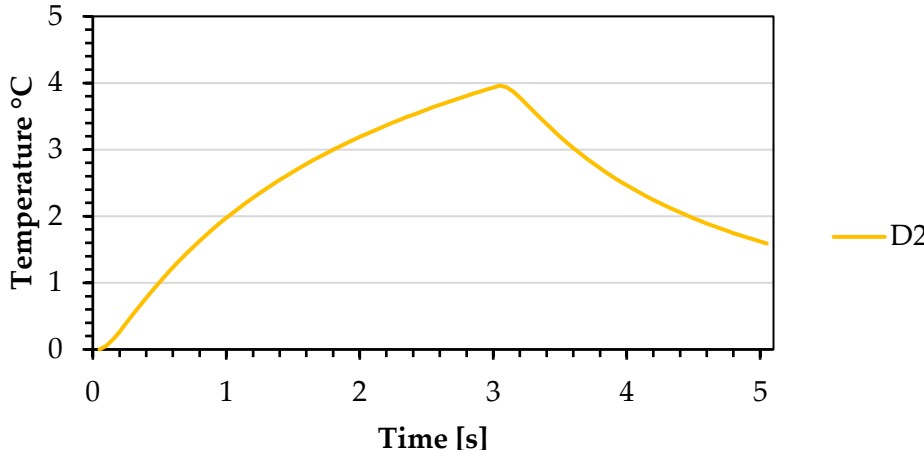

**Figure 10.** The course of temperature changes on the sample surface over the defects (location of the center of the vibration transducer x = 25, y = 55).

The presented simulation results clearly show that the impact on the detection of the defect (increase in the temperature value on the surface of the sample above the defect), with the same thermal forcing parameters (vibration transducer operating parameters), is significantly influenced by three factors: the size of the defect area, the distance between the defect, the transducer application point, and the depth under the surface where the defect is located. The defect D1, which has the largest surface area, generates the highest increase in temperature, as it can be seen in the graphs in Figures 7 and 9. Figure 7 shows the effect of the depth at which the defect is located under the sample surface. The transducer application point is at a similar distance from the D2 and D3 defects. The defect D2 is located deeper under the surface of the sample and the value of the temperature rise on this defect is lower than that on the defect D3, even though the surfaces of these defects are the same. The shift of the transducer application points towards the D2 defect (Figure 10), which resulted in a significant increase in the value of the temperature signal over this defect compared to the graph shown in Figure 7. However, there is no increase in the temperature signal over the D1 and D3 defects. This proves that the defect detection is possible with the distance of the transducer application not more than a few centimeters from the defect. No experimental tests have been carried out that would confirm the results of numerical calculations due to the lack of access to a source generating vibrations with parameters specified in the numerical calculations.

In order to compare the results with a different frequency of generated longitudinal waves but with the same other parameters, simulations were carried out for the frequencies of 20 kHz and 30 kHz. These are frequencies from the range of ultrasonic waves most often used for thermal excitation in ultrasonic thermography. The obtained results are presented in Table 3.

**Table 3.** The temperature increase over defects by thermal stimulation with ultrasounds.

| Defect Number | 20 kHz | 30 kHz |
| --- | --- | --- |
| D1 | 4.55 | 2.2 |
| D2 | 0.11 | 0.25 |
| D3 | 0.47 | 0.05 |

From the results presented in Table 3, it can be seen that the temperature rises over the defects when using a sound wave (Figure 3) is higher than when using ultrasound. This shows that thermal stimulation of the composite used to build boat hulls with the use of sound waves increases the probability of detecting a defect caused by the phenomenon of osmosis compared to ultrasonic waves.

### 5. Terahertz Test

As an alternative and future-proof method of non-destructive testing that can be used for damage testing of boat hulls, we present the terahertz transmission method.

As part of the work on the use of terahertz radiation in non-destructive testing of composites, experimental tests of the composite reinforced with glass fiber were also carried out. A sample of the composite approximately 10 mm thick with circular incomplete holes milled to various depths and diameters of different sizes was made at the Air Force Institute of Technology.

The transmission technique was chosen as the test method. The source of terahertz radiation and the scanner were stationary, and the test sample moved at a constant speed. Figure 11 shows a diagram of the stand.

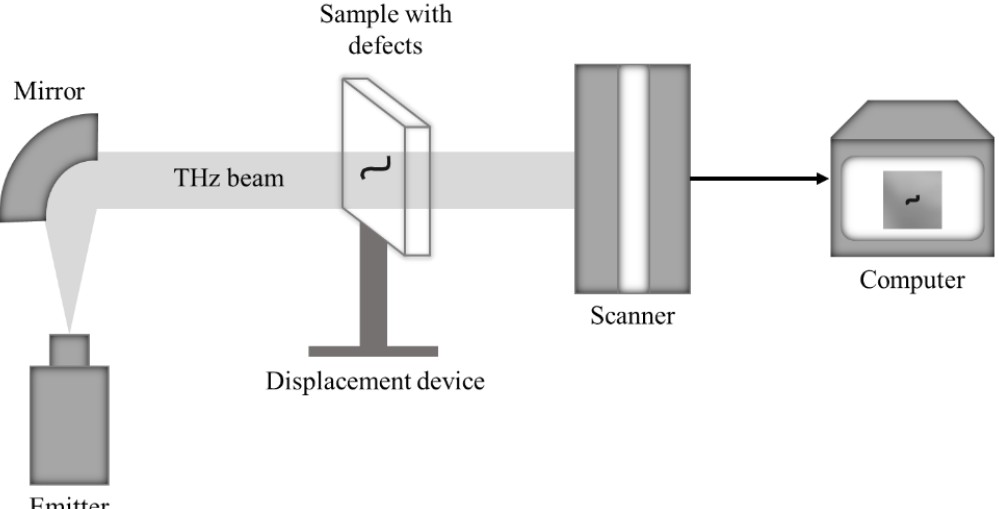

**Figure 11.** Schematic representation of experimental setup operated via Terahertz method in transmission mode [28].

Terasense Development Labs equipment was used in the study. A line scanner with the following parameters was used as a detector of terahertz radiation: image resolution 512 × 1 pixel with a pixel pitch of 0.5 mm and a frequency of ~300 GHz. The IMPATT THz generator, with the parameters of frequency 292 GHz $\pm$ 5 GHz and power ~10 mW, was the source of terahertz radiation. This source incorporates innovative THz reflective optics based on a specially configured high-gain horn antenna in combination with a metallic mirror. The distance between the radiation source antenna and the center of the mirror was 280 mm, and the distance between the center of the mirror and the scanner was 310 mm. The scanner was located about 5 mm from the sample surface.

Due to the adopted solution, the amount of power reaching the linear sensor matrix increases significantly. This has a decisive influence on the improvement of THz imaging. The data were recorded using dedicated software.

As can be seen in Figure 12, the dimensions and shapes of the incomplete holes are clearly visible. The outlines of the circles visible outside the periphery of the holes may be the result of terahertz radiation scattering through the walls of the holes or the detected damage resulting from hole milling. This phenomenon will be explained in the course of our further research.

In the THz image (Figure 12), disturbances (darker spots) are visible on the surface of the sample, apart from defects. These disturbances can be reduced by carefully selecting the appropriate sample speed.

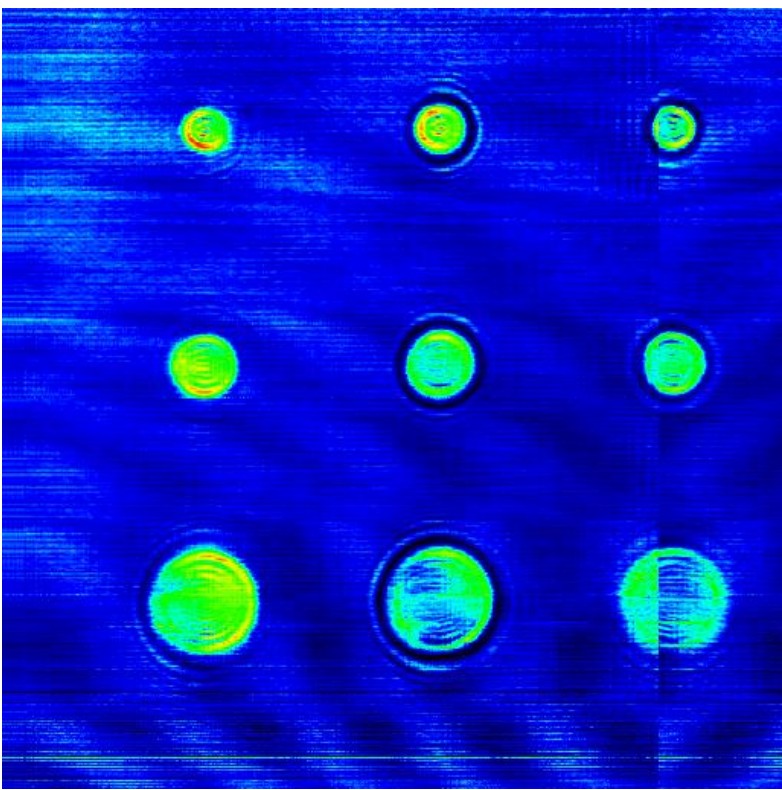

**Figure 12.** THz image of a glass–fiber reinforced composite with defects in the form of incomplete holes.

## 6. Conclusions

The computer simulations carried out have shown that vibration thermography can be an effective method in detecting defects in the hulls of vessels made of fiberglass-based composite. Both damages caused by the phenomenon of osmosis and impacts can be effectively detected by this method. The selection of the vibration source parameters is important.

In our simulations, we used sound waves that are unusual and very rarely used in vibrothermography, which (as our simulations have shown) can be more effective in detecting defects in boat hulls than the use of the ultrasound. The highest temperature increase over the defects was achieved at the frequency of 1 kHz (acoustic wave). This shows that at this frequency it will be possible to detect defects of much smaller dimensions that are located much deeper in the structure of the composite than at other frequencies.

In further work, we plan to conduct experimental research in order to verify our numerical calculations regarding the use of acoustic waves in the detection of defects caused by the phenomenon of osmosis. We intend to determine what the limitations are of the method resulting from the dimensions of the defects and at what stage of osmosis we can detect its destructive effects.

As results from the test carried out with the terahertz method show, it has great potential for its application to the detection of defects in composites. This technique is constantly evolving and has great potential in the future for non-destructive testing.

In further research, we will test this method in the reflection configuration, which is more practical due to the research object, which is the hull of the vessel. Both the radiation source and the scanner are located on the same side of the tested object. In this case, the hull will be stationary and the source with the scanner will move along the hull.

**Author Contributions:** Conceptualization, W.S.; methodology, W.S.; software, W.S.; formal analysis, M.S.; investigation, W.S.; resources, M.S.; data curation, M.S.; writing—original draft preparation, W.S.; writing—review and editing, M.S.; visualization, M.S.; supervision, W.S.; project administration, W.S.; funding acquisition, W.S. All authors have read and agreed to the published version of the manuscript.

**Funding:** This research was funded by The National Centre for Research and Development, grant number DOB-SZAFIR/02/A/001/01/2020".

**Institutional Review Board Statement:** Not applicable.

**Informed Consent Statement:** Not applicable.

**Data Availability Statement:** Not applicable.

**Acknowledgments:** We would like to thank the employees of the Air Force Institute of Technology for making a sample of the glass-fiber reinforced composite used in our research.

**Conflicts of Interest:** The authors declare no conflict of interest.

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
