# Peer review of "Possibilities of Detecting Damage Due to Osmosis of GFRP Composites Used in Marine Applications"

_applsci, doi:10.3390/app13074171_

Round 1

Reviewer 1 Report

1.Kindly include some finding (not more than 2 lines) in the abstract.

2. Include some recent article in introduction part.

3. Include some concise information in Conclusion

Author Response

We thank the reviewer for his comments and suggestions for changes in our article. Below are our responses to the reviewer's comments.

1.Kindly include some finding (not more than 2 lines) in the abstract.

 Answer: In accordance with the reviewers' comments, a number of changes were made to the abstract. Your suggestion for a change has also been taken into account.

  1. Include some recent article in introduction part.

 Answer: Two new articles related to the subject of the article have been added in the introduction.

  1. Include some concise information in Conclusion

 Answer: The comment was taken into account and the conclusions were extended.

Reviewer 2 Report

It is not an attractive manuscript since the authors did not present your idea clearly. From the abstract, the authors did not persuade readers of what is your main contribution and how interesting your results. Besides, a few words were repeatedly used without concentration on your work.

From the title of this manuscript, the authors seem to focus on the osmosis of GFRP. However, another defect of impact was also considered in this manuscript, why?

Please explain again the meaning of x+0, x-0, y+0, y-0, z+0, and z-0?

Please check the terminology again: shear elastic modulus?

Why did the authors use the equations from (1) to (3)?

What is the physical meaning and how to determine kfr, Sfr from eq. (4)?

From these equations, how did the authors develop your program to detect the defects? Compared to commercial software, what is your advanced?

In section 3.2, the authors mentioned the lateral size of 250 mm, but the real model was only 100 mm, why?

Please check again the grammar errors, typos, font size, figure quality, and labels of figures. I cannot see any number in Fig. 4, 6. Especially, Fig.  4 should add the notice to explain the difference of black, grey, and green colors.

Please describe in more detail the experiment, setting of equipment, and your main contribution to development of the detecting method?

Author Response

We thank the reviewer for his comments and suggestions for changes in our article. Below are our responses to the reviewer's comments.

  1. From the abstract, the authors did not persuade readers of what is your main contribution and how interesting your results. Besides, a few words were repeatedly used without concentration on your work.

Answer: The content of the abstract has been changed and expanded, and your comments have been taken into account.

  1. From the title of this manuscript, the authors seem to focus on the osmosis of GFRP. However, another defect of impact was also considered in this manuscript, why?

Answer: Of course, the article deals with the detection of defects due to the phenomenon of osmosis. We wrote about impact defects for a reason, because impact dents are the places where defects caused by osmosis occur in the first place. In addition, during the non-destructive testing that we propose for use, both types of defects are detected. Appropriate corrections regarding this issue have also been introduced in the article.

  1. Please explain again the meaning of x+0, x-0, y+0, y-0, z+0, and z-0?

Answer: x+0 specifies infinitesimal neighborhood to the right from the volume considered.  Respectively: x-0 is to the left.  y-0 means a point closer to the observer by the Y axis. Respectively, y+0 means a point farther from the observer. z+0 means higher by the Z axis, and z+0 means lower by the z axis.

  1. Please check the terminology again: shear elastic modulus?

Answer: Yes, we agree that it is better to use the term "shear modulus". In fact this modulus is a measure of the elastic shear stiffness of a material. Corrected in the article.

  1. Why did the authors use the equations from (1) to (3)?

Answer: Equations (1-3) are conventionally used to derive Lame equations, which describe deformation of elastic bodies.

  1. What is the physical meaning and how to determine kfr, Sfrfrom eq. (4)?

Answer: kfr is the friction coefficient. Sfr is the area subjected to friction. Both parameters are decisive for calculating a released energy.

  1. From these equations, how did the authors develop your program to detect the defects? Compared to commercial software, what is your advanced?

Answer: ThermoSon is a modeling software to predict magnitude of temperature signals because of friction in solids subjected to ultrasonic stimulation. It is fairly difficult to perform measurement of effective ultrasonic power acting within a body.  The program uses the finite difference method to solve the corresponding equations. The Program is less flexible but faster than the  finite element method. It is specialized and therefore more robust and easier to use than commercial software.

  1. In section 3.2, the authors mentioned the lateral size of 250 mm, but the real model was only 100 mm, why?

Answer: We would like to clarify that the sample dimension is correct. In Figure 5, the grid for numerical calculations is marked, not the sample size in mm.

  1. Please check again the grammar errors, typos, font size, figure quality, and labels of figures. I cannot see any number in Fig. 4, 6. Especially, Fig.  4 should add the notice to explain the difference of black, grey, and green colors.

Answer: The figures have been corrected. 

Please describe in more detail the experiment, setting of equipment, and your main contribution to development of the detecting method?

Answer: The description has been completed.

Reviewer 3 Report

Investigation of damage due to osmosis in GFRP using NDT needs improvement, including:

1. Research gaps and novelty have not been explained explicitly in the introduction

2. It is necessary to visualize the damage due to osmosis in marine applications

3. The purpose of Section 2 of vibrothermography is not clear, does it explain the method or explain the research design used? If you explain about vibrothermography, it is better if it is in the literature review section or explained in the introduction

4. The experimental model is explained more specifically in the Methodology Section

5. The image quality in Figure 4 is very poor and does not provide any information

6. Section Result and Discussion is not comprehensive. There is no explanation in Figure 5, Figure 6, Figure 7

7. Discussions need to be added more comprehensively and use relevant citations

8. Conclusions need to be written more scientifically and improved according to the title of the proposed study

Author Response

  1. Research gaps and novelty have not been explained explicitly in the introduction

Answer: The introduction has been revised taking into account the comments.

  1. It is necessary to visualize the damage due to osmosis in marine applications

Answer: A photo with visible effects of osmosis has been added to the article.

  1. The purpose of Section 2 of vibrothermography is not clear, does it explain the method or explain the research design used? If you explain about vibrothermography, it is better if it is in the literature review section or explained in the introduction

Answer: Since the main method that we intend to use in detecting osmotic defects is vibrothermography, we decided that it should be distinguished and presented in a separate section.

  1. The experimental model is explained more specifically in the Methodology Section

Answer: The model used for numerical calculations differs from the model used in experimental studies. We had completely different goals when studying these models that are described in the article.

  1. The image quality in Figure 4 is very poor and does not provide any information

Answer: Figure 5 has been corrected, it was previously numbered 4.

  1. Section Result and Discussion is not comprehensive. There is no explanation in Figure 5, Figure 6, Figure 7

Answer: The description of the results has been extended and supplemented.

  1. Discussions need to be added more comprehensively and use relevant citations

Answer: We have made corrections.

  1. Conclusions need to be written more scientifically and improved according to the title of the proposed study

Answer: Conclusions have been revised and expanded.

Round 2

Reviewer 2 Report

Agree with the revision and response

Author Response

Thank you for your help and comments regarding our article. We have also introduced a number of additions and corrections to the content of the article regarding: abstract, introduction, description of the vibrothermography method and conclusions.

Reviewer 3 Report

In fig 2, can you show a better resolution photo

Author Response

Thank you for your comments and suggestions for improvements:
The photo in Figure 2 has been changed and paper [28] has been added to the reference.